# Three-Dimensional Printed Polyamide 12 (PA12) and Polylactic Acid (PLA) Alumina (Al_2_O_3_) Nanocomposites with Significantly Enhanced Tensile, Flexural, and Impact Properties

**DOI:** 10.3390/nano12234292

**Published:** 2022-12-02

**Authors:** Markos Petousis, Nectarios Vidakis, Nikolaos Mountakis, Vassilis Papadakis, Lazaros Tzounis

**Affiliations:** 1Mechanical Engineering Department, Hellenic Mediterranean University, Estavromenos, 71004 Heraklion, Greece; 2Institute of Molecular Biology and Biotechnology, Foundation for Research and Technology—Hellas, 71110 Heraklion, Greece

**Keywords:** fused filament fabrication (FFF), 3D printing (3DP), polymer nanocomposites, nanoparticles (NPs), melt-processing, Polyamide 12 (PA12), Polylactic acid (PLA), additive manufacturing (AM), rapid prototyping, mechanical properties, aluminum oxide (Al_2_O_3_)

## Abstract

The effect of aluminum oxide (Al_2_O_3_) nanoparticles (NPs) as a reinforcing agent of Polyamide 12 (PA12) and Polylactic acid (PLA) in fused filament fabrication (FFF) three-dimensional printing (3DP) is reported herein for the first time. Alumina NPs are incorporated via a melt–mixing compounding process, at four different filler loadings. Neat as well as nanocomposite 3DP filaments are prepared as feedstock for the 3DP manufacturing of specimens which are thoroughly investigated for their mechanical properties. Thermogravimetric analyses (TGA) and Raman spectroscopy (RS) proved the nature of the materials. Their morphological characteristics were thoroughly investigated with scanning electron and atomic force microscopy. Al_2_O_3_ NPs exhibited a positive reinforcement mechanism at all filler loadings, while the mechanical percolation threshold with the maximum increase of performance was found between 1.0–2.0 wt.% filler loading (1.0 wt.% for PA12, 41.1%, and 56.4% increase in strength and modulus, respectively; 2.0 wt.% for PLA, 40.2%, and 27.1% increase in strength and modulus, respectively). The combination of 3DP and polymer engineering using nanocomposite PA12 and PLA filaments with low-cost filler additives, e.g., Al_2_O_3_ NPs, could open new avenues towards a series of potential applications using thermoplastic engineering polymers in FFF 3DP manufacturing.

## 1. Introduction

Three-dimensional (3D) printing (3DP), which has evolved over the last three decades from the initial reporting and is defined as rapid prototyping, is nowadays considered as one of the most promising and disruptive manufacturing technologies [1,2]. More specifically, 3DP is an additive manufacturing (AM) technology, in which 3D objects consisting of different materials, e.g., metals [3,4], ceramics [5,6], polymers, i.e., thermoplastics [7], thermosets [8] and elastomers [9], and composites [10] are constructed in a sequential layer-by-layer manufacturing approach directly from computer-aided design (CAD) models. As such, 3D objects consisting of different materials [11,12], complex geometries [13,14], and by-design and tailored final component bulk properties, i.e., mechanical [15], thermal [16], electrical [17], magnetic [18], catalytic [19], antimicrobial [20], etc., could be easily realized due to the unique nature of 3DP, compared to traditional manufacturing methods (injection molding, plastic forming, CNC machining, etc.) To that end, several high-end applications have been reported; namely, in the biomedical field [21], e.g., implants [22], scaffolds for tissue engineering [23], surgical equipment [24], medical diagnostic tools [25], tissue and organ printing [26], etc., up to 4D printed structures for robotics applications [27,28], all employing 3DP as the main manufacturing and fabrication technology alternative to traditional manufacturing methods.

It was back in 1982 when Hideo Kodama reported the first 3D printed part [29]. Since then, there have been reported various 3DP technologies for the manufacturing of polymers, metals, ceramics, and composite parts with variable shapes and properties, arising both from the selection of the materials used as feedstock, as well as the proper 3DP AM process parameters [30]. There are various 3DP AM technologies; for instance: (i) Binder Jetting, (ii) Selective Laser Sintering (SLS), e.g., direct metal laser sintering and selective laser sintering of thermoplastic polymeric powders, (iii) Stereolithography, e.g., using photopolymerizable polymeric materials, (iv) Fused Filament Fabrication (FFF), (v) Digital Light processing (DLP), and (vi) Material Jetting and Drop on Demand (DOD) [31]. Amongst others, in the FFF process, a thermoplastic polymeric filament is heated up to temperatures higher than its melting point (T_m_). At this state, the polymeric material is extruded from a nozzle that moves in XYZ space. FFF has received continuous scientific and technological interest, especially when it comes to the manufacturing of thermoplastic polymeric material components [32,33] and/or polymer composites, e.g., nanocomposites [34,35] and fiber-reinforced composites [36,37]. To date, several achievements have been made on the level of the 3D printer as a machine; namely, the printer’s accuracy, automation, printer parts, e.g., nozzles, heat-able beds, etc., while all could affect the quality of the final manufactured parts [38]. However, thermoplastic materials with “niche” and/or enhanced mechanical, magnetic, thermal, and electrical properties compared to pure polymer is still a field of ongoing research to endow a multi-functional character to the final printed parts [39].

Poly-Lactic Acid (PLA) is a biocompatible and biodegradable polyester thermoplastic in nature with unique mechanical strength and ease of melt-processing, making it thus one of the most widely-used feedstock materials in FFF 3DP technology [40,41]. PLA has a relatively high melting point (T_m_ in the range of ~150–160 °C), which makes it a promising candidate material for engineering applications, apart from the well-known and various biomedical applications. Polyamide 12 (PA12) is an “engineering” thermoplastic as well, with high potential for 3DP applications, especially in the FFF process, belonging to the huge family of polyamides known for their unique toughness, strength, impact, and resistance to fracture upon being exposed to a great level of deformation [42]. In general, PA12 has been extensively used in SLS 3DP processes already back in 2010 [43] and is not so commonly used in FFF apart from recent studies [17,42,44]. On the other hand, its market price together with the combined physicochemical properties and the ease of processing in 3DP could make it an ideal material for rapid prototyping purposes up to advanced applications, e.g., structural parts with unique toughness properties. The literature review on PA12 and PLA in FFF 3D printing shows the potential of the polymers in the process and the focus of the research about them.

Polymeric materials [45], as well as polymer nanocomposites, have lately received considerable attention in FFF 3DP [46], since they could offer a facile approach to improving the 3DP part’s properties; for instance, (i) the quasi-static, dynamic, and thermomechanical properties of the host polymeric matrix [47], (ii) nano-inclusions could induce crystallization of the polymer matrix, especially for semi-crystalline polymers, with the nanofillers functioning as nucleating agents, which can have a positive effect then on the mechanical properties, the thermal stability, etc. [48], and (iii) 3DP parts with multi-functional properties, e.g., sensing, electrical, optical, actuation, etc., can be realized [49]. Nanoparticles (NPs) with different geometries, i.e., 0D: spherical, e.g., SiO_2_, ZnO, Al_2_O_3_ NPs; 1D: wires or tubes, e.g., carbon nanotubes (CNTs); 2D: platelet-like, e.g., graphene, clays, etc., and at different loadings have been incorporated via melt–mixing processes in thermoplastic polymeric filaments used further for FFF 3DP. Carbon nanotubes (CNTs) and Graphene (Gr) have been used to enhance the mechanical properties as well as endow electrically conductive 3DP parts [50]. These works highlight the effect of the nanoparticles’ addition as fillers in polymer matrices, which enhance and expand their properties, making them suitable for a wider range of industrial applications.

Research on aluminum oxide (Al_2_O_3_) has been presented on phase composition, chemical composition, and morphology of the powder [51]. The behavior of alumina as a filler has also been investigated, aiming to improve the dispersion in the composites and the adhesion in 3D printing processes [52,53,54,55]. Alumina NPs have numerous physicochemical properties, e.g., optical transmission, electrical insulators, chemically inert, they maintain their thermal stability even at increased temperatures, etc. Additionally, alumina NPs are a nano-additive often used in the industry of thermoplastics [56]. Relatively, the use of Al_2_O_3_ NPs in 3DP polypropylene (PP)/Al_2_O_3_ nanocomposites, with the tensile and flexural strength to be increased by approx. 4% and 19% for the PP/Al_2_O_3_ (1.0 wt.%) nanocomposite has been recently reported. Alumina has been used as filler in composites with polyamides [57]. In this work, alumina was not in nanoparticle form. The composites were developed using a solvothermal method and composites were not compatible with the 3D printing process. The study reports a reinforcement in the mechanical properties of the composites; still, it focuses mainly on the thermal and electrical properties of the developed composites. In another study in the literature, polyamide with alumina has been investigated in coatings [58]. AM PA12 with alumina composites has been investigated so far for biomedical applications [59]. A chemical process was followed for the preparation of the composites, which did not use NPs as fillers. Specifically, this work focused on investigating the effect of artificial saliva on the tensile properties of the composites. The effect of aging and sterilization on their physicochemical properties has also been reported [60].

Research in nanocomposites of PLA with alumina is focused on the thermal and electrical properties of the composites [61], but not on 3D printing as a manufacturing process. PLA in fiber form has also been used for the preparation of composites with alumina, employing the electrospinning process [62]. Poly(butylene adipate-co-terephthalate) and alumina nanoparticles have been combined for the reinforcement of PLA but not in 3D printing applications [63]. In the blends developed in the work, the results were not very promising in the flexural tests due to the poor dispersion of the alumina in the blends, while the impact strength was significantly increased. In film applications, alumina has been introduced in the PLA polymer, aiming to improve the optical, morphological, structural, and mechanical properties of the polymer [64]. In 3D printing, composites using PLA as the matrix material and alumina as filler combined with graphene have been presented, focusing on the thermal conductivity of the developed composites [65]. This work did not investigate the mechanical properties of the composites, which were prepared with a process compatible with the laser sintering AM procedure. To date, there is no study having been reported on the utilization of Al_2_O_3_ NPs as a filler for mechanical reinforcement of 3D printed PA12 and PLA thermoplastic materials, nor an in-depth study of the mechanical and fracture properties of the manufactured 3DP specimens. The motivation herein was to develop nano-compounds in a form compatible with the FFF 3D printing process, with an improved mechanical response, aiming to expand the fields of use of the process, in areas in which superior mechanical response from the build parts is the required specification. Additionally, a prerequisite in the work was to follow an industrial-ready procedure, without significantly increasing the overall cost of the materials and the preparation procedure.

In this study, for the first time, two different thermoplastics in nature and polar semi-crystalline polymeric materials, namely PA12 and PLA, with different macromolecular architectures and side functional groups have been reinforced with Al_2_O_3_ NPs inorganic low-cost fillers towards the production of nanocomposite filaments used to manufacture 3DP nanocomposite specimens. The reinforcement mechanism has been compared for the two polymeric matrices, with the PA12 showing a slightly more pronounced mechanical properties’ enhancement, while the filler loading was kept constant at 1.0, 2.0, 3.0, and 4.0 wt.% in both cases to understand the basic process-structure-property relationship for PA12 and PLA Al_2_O_3_ nanocomposites.

## 2. Materials and Methods

### 2.1. Materials Used in the Work for the Preparation of the Nano-Compounds

Polyamide 12 (PA12) of AESNO TL grade (Arkema S.A., Rilsamid PA12 AESNO TL, Colombes, France) in fine granules form was purchased from Arkema (Colombes, France). PA12 has a Melt Volume-Flow Rate (MVR) of 8.0 cm^3^/10 min at 235 °C/5.0 kg, Vicat Softening Temperature at 142 °C (according to ISO 306/B50), Melting Temperature of 180 °C (according to ISO 11357-3), and density of 1.01 g/cm^2^ (according to ISO 1183) according to the supplier’s specifications. Polylactic Acid (PLA) in the form of coarse powder as a multi-functional thermoplastic polymer, used in various engineering as well as bio-related applications and is a biocompatible and biodegradable polymer, was received from Plastika Kritis SA (Heraklion, Crete, Greece) with a commercial batch name 3052D grade and a molecular weight of 116.000 g/mol. Both polymer grades used throughout this study as the matrix material for the fabricated nanocomposites are appropriate for melt mixing processes and contain heat stabilizing, lubrication, and UV stabilizer additives, according to the suppliers’ technical data sheets. The aluminum oxide (Al_2_O_3_) nanoparticles (NPs) that have been used as mechanical reinforcement blended fillers, both for PA12 and PLA matrices in this study, are spherical in shape with an average NP diameter of ~180 nm (grade NG04SO103) and procured from Nanographi (Nanografi Inc., Ankara, Turkey).

### 2.2. Preparation of the Filament and Process Parameters for FFF 3D Printing of the PA12, PLA, with Al_2_O_3_ Filler Nanocomposites

In the work, nanocomposites were prepared with increasing loading. Specimens were fabricated and tested, and the results were evaluated. Then, the filler loading was increased. When the mechanical properties of a specific loading started to decline, the experimental course was terminated. Nanocomposites were prepared from the beginning with the same process and material grades. All the tests were conducted with the same conditions to have comparable results.

The reinforcement effect of the alumina additive was investigated in the two thermoplastics. Additionally, another criterion was to test low filler concentrations which would not change other parameters and aspects of the polymers significantly, such as their rheology. At the same time, higher filler concentrations would have an effect on the cost of the produced nanocomposites. A threshold analysis was carried out in the work for the reinforcing effect of alumina in the two polymers, keeping the other parameters as close as possible to the properties of the pure polymers.

Following this analysis and specifications, initially, PA12 and PLA as received thermoplastic materials were physically mixed in raw powder form with 1.0, 2.0, 3.0, and 4.0 wt.% of Al_2_O_3_ NPs solid filler content, respectively, using a mechanical homogenizer. With this pre-compounding process, raw materials were mixed at the predefined concentration ratios mentioned above and different mixtures were prepared for each nano-compound studied in the work (polymer and filler loading).

Predetermined quantities for each nano-compound were dried at 80 °C in an oven overnight prior to the filament extrusion process. Hereafter, the PA12/Al_2_O_3_ and PLA/Al_2_O_3_ nanocomposites, namely the 3DP nanocomposites extruded as filaments or as FFF fabricated printed samples, are denoted as PA12/Al_2_O_3_ (1.0 wt.%), PA12/Al_2_O_3_ (2.0 wt.%), PA12/Al_2_O_3_ (3.0 wt.%), PA12/Al_2_O_3_ (4.0 wt.%), and PLA/Al_2_O_3_ (1.0 wt.%), PLA/Al_2_O_3_ (2.0 wt.%), PLA/Al_2_O_3_ (3.0 wt.%), PLA/Al_2_O_3_ (4.0 wt.%) for the 1.0, 2.0, 3.0, and 4.0 wt.% different additive concentration, respectively.

Raw materials were first converted into 3DP filaments. A two-step process was followed to achieve as good dispersion of the alumina additive in the matrices as possible. Raw materials in powder form were fed into a Noztek extruder (Noztek, Shoreham-by-Sea, UK) for the initial mixing of the raw materials. The filament was shredded into pellets (3 devo shredder, 3 devo, Utrecht, The Netherlands). The pellets were then used in an additional extrusion process for the manufacturing of the 3DP filament with a 1.75 mm diameter, suitable for FFF (3D Evo Composer 450 materials mixing extruder featuring a single screw, 3D Evo B.V., Utrecht, Netherlands).

Initial tests both for the PA12 and PLA matrix materials have been performed to optimize the filament extrusion parameters, namely, (i) mixing speed, (ii) appropriate screw for the mixing/compounding process, and (iii) extrusion temperatures. More specifically, the 3D Evo Composer 450 has four (4) heating zones through the overall extrusion path/chamber, while the 3DP filament diameter is monitored continuously with an in-built sensor within an acceptable range of 1.68 mm ± 0.07 mm, to adjust the extrusion speed in-line with the extrusion process. As such, one can achieve a high quality of the produced filament with real-time filament diameter metrology, guaranteeing thus the filament diameter accuracy across its length. Neat thermoplastic PA12 and PA12/Al_2_O_3_, as well as neat PLA and PLA/Al_2_O_3_ 3DP filaments, were processed under the same extrusion parameters, respectively. Specifically, for neat PA12 and PA12/Al_2_O_3_ 3DP filaments, the following temperature was set for each heating zone: close to the nozzle—210 °C, middle zones—220 °C, and close to the hopper—185 °C. For neat PLA and PLA/Al_2_O_3_ 3DP filaments, the parameters, respectively, have been: 195 °C, 205 °C (middle stage), and 175 °C close to the hopper. In both cases, the screw rotational speed was set to 8.5 rpm and the machine’s built-in rewinder was employed to rotational speeds appropriate to achieve the required filament diameters for all the different systems (3devo Composer 450 operates in the range of 3–15 rpm). Moreover, an extra air duct of the machine directly after the extruder’s nozzle was employed to support the filament’s cooling procedure, having a direct effect on the roundness of the filament. It is worth mentioning that high quality of filament, i.e., in diameter homogeneity, roundness, dispersion of nanoparticles, etc., is a prerequisite towards consistent and high-quality FFF 3DP objects. All produced filaments were dried at 80 °C overnight, before being used to manufacture the different 3DP specimens.

The FFF 3DP process manufactured the pure thermoplastic PA12 and the PA12/Al_2_O_3_ specimens as well as neat PLA and PLA/Al_2_O_3_ 3DP specimens using an Intamsys 3D Printer (Funmat HT 3D FFF technology, Intamsys Technology Co. Ltd., Shanghai, China). In the beginning, for neat PA12 and PLA polymeric matrices, as well as for the polymer nanocomposite filaments, 3DP trials were performed using different parameters to determine the optimum set of 3DP experimental conditions yielding high-quality 3DP specimens. Finally, for PA12 and PA12/Al_2_O_3_ nanocomposites the following parameters were used: 100% solid infill, 40 mm/s print speed, 45 degrees deposition orientation angle (raster angle), 0.20 mm layer height, 270 °C nozzle temperature, and 90 °C bed temperature, while for PLA and PLA/Al_2_O_3_ nanocomposites: 100% solid infill, 40 mm/s print speed, 45 degrees deposition orientation angle, a layer thickness of 0.20 mm, 210 °C nozzle temperature, and 50 °C bed temperature. Most of the filament extrusion and 3DP parameters for PA12 [44] and PLA [66], especially that of the neat polymeric materials, have been developed in our previous studies.

Figure 1 depicts schematically the steps of the methodology implemented in this research article to manufacture the different specimens, i.e., starting from raw materials towards the preparation of 3DP filaments, the 3DP process of the different specimens, as well as finally the mechanical and morphological characterization analyses. Figure 2 summarizes the optimum 3DP parameters for the different systems to fabricate specimens, the dimensions of the 3DP specimens used for different mechanical characterization techniques, and the real 3DP manufactured specimens (representative samples for tensile, flexural, and impact tests).

### 2.3. Characterization Techniques

Raman investigations were performed with an LabRAM HR Raman Spectrometer (HORIBA Scientific, Kyoto, Japan). A laser beam with an excitation wavelength of 532 nm and a maximum output power of 90 mW was employed. A 50× objective with 0.5 numerical aperture and 10.6 mm working distance (LMPlanFL N, Olympus, Tokyo, Japan) delivered the excitation light and collected the Raman activity. For each spectrum, the spectral was set at 40 to 3900 cm^−1^, with an acquisition time of 10 s and 5 accumulations.

Thermogravimetric analysis (TGA) in a nitrogen atmosphere was carried out for PA12, and PLA as well as for the PA12/Al_2_O_3_ and PLA/Al_2_O_3_ nanocomposite 3DP specimens. TGA measurements were conducted employing a Perkin Elmer Diamond TG/TDA (Waltham, MA, USA) apparatus with a heating pattern from 30 °C to 550 °C and a step of 10 °C/min.

Tapping Mode Atomic Force Microscopy (AFM) was carried out using a MicroscopeSolver P47H Pro scanning probe microscope (NT-MDT, Moscow, Russia) in ambient conditions. Commercially available silicon cantilevers were used with a tip radius of about 10 nm, a tip cone angle of 20°, and a cantilever spring constant of 35 N/m at a scanning frequency of 1 Hz. The surface roughness values of all extruded filaments used for 3DP were determined after the 2nd flattening operation from the overall captured area of 10.0 × 10.0 μm^2^ of the corresponding samples’ height images.

The microstructure of the side surface of the 3DP specimens as well as the cracked surfaces in the tensile experiments on the 3DP specimens was examined using SEM characterization by a Jeol JSM-IT700HR (Jeol Ltd., Tokyo, Japan) field emission SEM in high-vacuum mode at 20 kV acceleration voltage. A Secondary Electron (SE) detector was used for the observation of the samples. To avoid charging effects, samples were Au sputter coated (5 nm thin film). Energy Dispersive X-ray Analysis (EDS) has been performed for Al_2_O_3_ NPs without a sputtered Au layer.

Tensile tests were conducted according to the ASTM D638-02a international standard. All tests were performed at a room temperature of 23 °C. Three-point bending flexural tests were conducted following the ASTM D790-10 international standard. 3DP specimens had dimensions of 64.0 mm length, 12.4 mm width, and 3.2 mm thickness, and the support span was set to 52.0 mm. An Imada MX2 (Imada inc., Northbrook, IL, USA) machine in tension (elongation rate of 10 mm/min) and flexural mode setup (testing speed: 10 mm/min), respectively, were employed for the tensile and three-point bending tests. Impact tests were carried out following the ASTM D6110-04 international standard. Notched specimens had dimensions of 80.0 mm (length) × 8.0 mm (width) × 10.0 mm (thickness). A Terco MT 220 (Terco, Huddinge, Sweden) Charpy’s impact apparatus was employed in the tests. For all the mechanical tests performed in this study, six (6) specimens have been assessed for pure as well as PA12/Al_2_O_3_ and PLA/Al_2_O_3_ 3DP nanocomposites, and the average values together with the corresponding standard deviations are reported. Moreover, the same 3DP parameters have been used to manufacture the tensile, flexural, and impact test specimens. Finally, as a part of the mechanical test campaign, microhardness measurements were conducted according to the ASTM E384-17. The 3DP specimens’ surface was thoroughly polished before each measurement. An Innova Test 300-Vickers (Innovatest Europe BV, Maastricht, The Netherlands) testing machine was employed, while the applied force was set to 100 gF, and a duration of 10 s was selected for the indentation. Imprints were measured for six (6) different specimens of the neat PA12 and PLA, as well as the PA12/Al_2_O_3_ and PLA/Al_2_O_3_ nanocomposites. All parameters of the mechanical test campaign have opted for 3D specimens based on our previous detailed parametric study [67].

## 3. Results and Discussion

### 3.1. Raman and EDS Analysis of Neat PA12, PLA, PA12/Al_2_O_3_, and PLA/Al_2_O_3_ Nanocomposites

In Figure 3A, the major Raman peaks arising from PA12 are presented, while no significant differences were observed between the PA12 and the samples with Al_2_O_3_ nanoparticles. Clearly, C–O–C stretching was identified at 1060, 1105, and 1293 cm^−1^. CH_2_ and CH_2_ deformations were identified at 1418 and 1441 cm^−1^, respectively. Lastly, CH_2_ symmetric stretching and deformation were identified at 1434 cm^−1^, 2850 cm^−1^, 2884 cm^−1^, and 2923 cm^−1^ (Table 1). As is seen in Figure 4A, the major Raman peaks are due to PLA. The addition of Al_2_O_3_ NPs in PLA showed no significant changes in the Raman spectrum. From the analysis of neat PLA, the major Raman peaks were identified, and their related assignments are presented in Table 2. The range of the Raman peaks found is between 870 cm^−1^ and 2996 cm^−1^. EDS spectra for all 3DP PA12/Al_2_O_3_ and PLA/Al_2_O_3_ nanocomposites showing the existence of atomic Al arising from the Al_2_O_3_ incorporated nanoparticles in the PA12 and PLA polymer matrix are shown in Figure 3B–E and Figure 4B–E, respectively, for all the different filler loadings.

### 3.2. TGA Analysis

TGA investigations have been performed to reveal the temperature stability of the different materials in this study under a nitrogen atmosphere [74]. Moreover, TGA graphs proved indirectly the nature of the different nanocomposites and the existence of the specific filler loading in each specimen, via the observed remnant material at temperatures where the polymer matrix (PA12 or PLA) has been totally decomposed. Figure 5 shows the TGA (Figure 5A), and DTG (Figure 5B) curves of 3DP PA12, PA12/Al_2_O_3_ (1.0 wt.%), PA12/Al_2_O_3_ (2.0 wt.%), PA12/Al_2_O_3_ (3.0 wt.%), PA12/Al_2_O_3_ (4.0 wt.%), and PLA, PLA/Al_2_O_3_ (1.0 wt.%), PLA/Al_2_O_3_ (2.0 wt.%), PLA/Al_2_O_3_ (3.0 wt.%), PLA/Al_2_O_3_ (4.0 wt.%). As it can be observed, PA12 is more thermally stable compared to PLA, with the onset temperature of decomposition ($T_{on}^{d}$) at 420 °C, whereas PLA is at 330 °C. At temperatures above 500 °C, both polymeric materials have been fully decomposed, and the remnant material in some curves corresponds to the Al_2_O_3_ nanofillers. The remaining mass after the completion of the TGA agrees with all nano-compounds studied herein.

From the insets in the TGA and DTG curves representing a small temperature window of the whole thermogram, it can be deduced that the addition of Al_2_O_3_ nanofillers slightly increased the thermal stability of both polymeric matrices, since the $T_{on}^{d}$ has been shifted to slightly higher temperatures. It can be deduced thus that the thermal stability of the polymer nanocomposites has been slightly improved by the existence of Al_2_O_3_ nanoparticles. Regarding the DTG graphs, a slightly different response is reported. In the PLA polymer, the addition of alumina shifts the highest weight loss ratio to slightly lower temperatures, and the rate is increased. On the other hand, the alumina introduction in the PA12 thermoplastic shifts the highest weight loss ratio to slightly higher temperatures, and the rate is slightly decreased. Still, differences are not significant and can be attributed to the different interactions of the additive with the different polymers. Finally, TGA and DTG analyses proved that the temperatures selected in the study for polymer processing, i.e., either the 3DP filament melt mixing and extrusion processes or the 3DP filamentous FFF 3DP manufacturing, are far below the decomposition temperatures of the neat polymeric matrices (PA12 and PLA).

### 3.3. 3DP Filament Diameter Optical Metrology

Figure 6 depicts the real-time monitored filament diameter of neat polymer matrices (PA12 and PLA), as well as for the two highly loaded nanocomposites, namely the PA12/Al_2_O_3_ (4.0 wt.%) and PLA/Al_2_O_3_ (4.0 wt.%) during a period of 30 min total extrusion time. It is well-known that for FFF 3DP manufacturing, an “almost” industry standard is to work with filaments exhibiting a 1.75 mm diameter. In the study, the 3D Evo Composer 450 single screw extruder has been employed (3D Evo B.V., Utrecht, The Netherlands) with an in-built sensor that measures the filament diameter continuously during the extrusion process. It can be observed that in all cases presented in Figure 6, the filament diameter is within an acceptable range of 1.68 mm ± 0.07 mm, which is achieved by the extruder’s function to adjust the extrusion speed in-line with the extrusion process to achieve such a constant filament diameter across its length maintaining and thus the aforementioned accuracy tolerances. It is worth mentioning that high quality of filament, i.e., in diameter homogeneity, roundness, dispersion of nanoparticles, etc., is a prerequisite towards consistent and high-quality 3DP objects. Moreover, it is of utmost importance to have filaments with a constant diameter. The measured diameter of the filament is a parameter considered by the slicer software when defining the 3DP process. The microscope images from the side surface of the filament, in all nano-compounds prepared in the work, reveal a smooth, defect and void-free surface, indicating a good quality filament, showing that the parameters used in the work were the appropriate ones.

### 3.4. AFM Surface Roughness Analysis of Neat Polymer and Nanocomposites 3DP Filaments

Figure 7 and Figure 8 show the 3D AFM topography images captured from the different 3DP extruded filaments in this study, utilized finally for the FFF 3DP manufacturing process of specimens for mechanical characterization. Specifically, Figure 7 depicts the topography images together with the respective roughness values (Rq: root-mean-square roughness, Ra: average roughness of a surface, Rz: the difference between the tallest “peak” and the deepest “valley” in the surface) of PA12, PA12/Al_2_O_3_ (1.0 wt.%), PA12/Al_2_O_3_ (2.0 wt.%), PA12/Al_2_O_3_ (3.0 wt.%), and PA12/Al_2_O_3_ (4.0 wt.%). On the other hand, the topography for PLA, PLA/Al_2_O_3_ (1.0 wt.%), PLA/Al_2_O_3_ (2.0 wt.%), PLA/Al_2_O_3_ (3.0 wt.%), and PLA/Al_2_O_3_ (4.0 wt.%) filaments are shown in Figure 8. A general observation is that in all cases, the surface roughness of filaments increases with the addition and increased amount of Al_2_O_3_ nanoparticles in the respective polymer matrix; however, it remains that the nanoscale is an important characteristic for the filament quality which can affect the consecutive 3DP process. The increase in surface roughness could be attributed (i) to some plausible existence of nanoparticles onto the filament surface, and (ii) PA12 and PLA polymer chains’ different conformation from the neat polymeric material, e.g., probably due to some polymer crystal formation caused by the incorporated Al_2_O_3_ nanoparticles.

The addition of the Al_2_O_3_ filler has a different effect on the surface roughness of the nanocomposites. Although PA12 pure has lower surface roughness than PLA pure, the nanocomposites having PA12 as the matrix material have higher surface roughness values than the corresponding PLA ones. The measured surface roughness values are very low, and the differences overall are not significant. Any differences can be attributed to the different rheological properties of the studied thermoplastics, which affect the surface structure of the materials. The addition of the aluminum oxide filler has a different effect on the structure of each thermoplastic. Additionally, measurements were taken at random positions, so differences are expected, also due to the topography of the microscale region the measurements were taken.

### 3.5. Scanning Electron Microscopy (SEM) Analysis of Al_2_O_3_ NPs and the Side Surface of the 3D Printed Specimens

In Figure 9, Al_2_O_3_ powder SEM images at two different magnifications (Figure 9A,B), together with the EDS spectrum (Figure 9C) are shown. The nature and nano-dimensions of particles are clearly observed with the spherical shape of the Al_2_O_3_ nanocrystallites. In the EDS analysis, the Al peak dominates the spectrum, as expected, verifying the existence of the Al element in the nanofiller (Figure 9C). The size of the NPs were verified in the SEM images. The Carbon element is expected in the EDS graphs for the 3D printed samples since they are made with polymeric materials, which are organic materials [75]. On the EDS graph for the alumina powder (Figure 9C), the powder is placed in a carbon tape to be inspected with SEM; therefore, the presence of carbon is expected in the graph, even though it is not an element of alumina. It should be mentioned that EDS is an indicative process regarding the presence of the elements; it is not for precise stoichiometric ratio or precise quantification of the concentration of an element since measurements are taken in a small region. High peaks are indicative of a high concentration of an element in the region under observation, but this is mainly for qualitative assessment, not an accurate quantitative one. Therefore, the expected exact ratio for Al and O in the alumina could not be reliably determined with the EDS results since additional measurements should have been taken.

Figure 10 and Figure 11 show the side surface morphology of all 3DP PA12/Al_2_O_3_ and PLA/Al_2_O_3_ nanocomposites, respectively, indirectly highlighting the 3DP specimens’ external structure, which is the result of the additively deposited layers and the underlying interlayer fusion. In Figure 10, two different magnifications, i.e., 30× and 150× are depicted, while in Figure 11 the side morphology of the samples is presented at 25× and 150× magnification levels. For all specimens, an excellent build structure and fusion between the layers could be observed, which highlights the high quality of the produced feedstock 3DP filaments, as well as the optimum 3D printing parameters selected for the specimens’ manufacturing in this research work. Moreover, the high quality of interlayer fusion without, e.g., discontinuities, voids, cracks, etc., could produce parts with layers that have strong interfacial shear strength, thus producing 3DP parts with a superior mechanical response. The defect-free interlayer fusion arises also from the fact that the alumina nanofillers have been most likely efficiently, and at the “nano-level”, dispersed in the polymer matrix. In any other case, they might have created some micro-aggregates impeding the polymer chain interdiffusion at the interphase between the layers, resulting thus in a microscopically observed deteriorated interlayer fusion. Moreover, this effect could have been caused even by the 3DP process, since micro-aggregates could result in the nozzle clogging being responsible for the quality of the final 3DP part via introducing inhomogeneities, defects, discontinuities, etc., all plausibly observed by SEM side surface morphology analyses. Only in the case of PA12/Al_2_O_3_ (3.0 wt.%), and (g, h) PA12/Al_2_O_3_ (4.0 wt.%) 3DP nanocomposites, some non-homogeneous in surface morphology side filaments and printed layer thickness could be observed, which is most likely attributed to an increased polymer melt viscosity affecting the 3D printing filamentous extrusion process slightly.

### 3.6. Tensile Properties of Filaments and 3DP Specimens: Neat PA12, PLA, and Their Al_2_O_3_ Nanocomposites

Tensile tests have been performed for the neat PA12, PLA, and the nanocomposite filaments at the first level (Figure 12), as well as on 3DP dog-bone tensile test specimens in the second stage (Figure 12). Figure 12A shows a representative neat polymer filament under testing, while Figure 12B is a nanocomposite filament. In Figure 12C,D, the tensile strength and the modulus of elasticity properties are summarized, respectively, for all the different extruded and produced filaments in this research work (mean values together with the corresponding standard deviation are shown). For all extruded filaments, Al_2_O_3_ NPs had a positive reinforcement effect for the different filler loadings. Namely, the highest increase was observed in tensile strength at 1.0 wt.% for PA12 (34.1%), and PLA (49.3%), while for the tensile modulus at 2.0 wt.% for PA12 (66.8%) and PLA (51.7%).

A rather similar pattern is observed in the two polymers, with an increase in the tensile strength at the 1 wt.% concentration and a decreasing trend with the further increase of the alumina filler concentration in the nano-compounds. Still, even at the higher loading of 4 wt.%, the tensile strength is higher than the pure material for both polymers. The reinforcement is significantly more intense in the PLA polymer than in the PA12 polymer.

Figure 13A,B present the comparative and representative Tensile stress (MPa) vs. strain (%) curves for PA12 and PA12/Al_2_O_3_ nanocomposites, as well as for PLA and PLA/Al_2_O_3_ nanocomposites, respectively (all curves for 3DP dog-bone shaped specimens described by the specific ASTM, with all details given in the Experimental section). As previously described for the extruded filaments, in Figure 13C,D, the tensile strength properties and modulus of elasticity values are summarized, for all the different 3DP dog-bone specimens (mean values together with the corresponding standard deviation). For all the 3DP specimens, Al_2_O_3_ NPs also exhibited a positive reinforcement effect for the different filler loadings. Especially, the highest increase has been experimentally determined for tensile strength and modulus at 1.0 wt.% for PA12 (41.1% and 56.4%, respectively), while at 2.0 wt.% for PLA (40.2% and 27.1%, respectively). The stress–strain curves of the pure 3DP polymeric materials coincide with the characteristic behavior observed in our previous study, where meticulous work was performed to identify the effect of the strain rate on the tensile properties (strength-stiffness) of different thermoplastic materials widely used in FFF 3DP [76].

From the above analyses, it can be easily realized that Al_2_O_3_ NPs have a more prominent reinforcement mechanism for tensile strength and modulus of elasticity at 1.0 and 2.0 wt.%, both for the extruded filaments, as well as for the respective 3DP specimens. On the other hand, there has been a marginal improvement for the 3.0 and 4.0 wt.% filler loadings, which most likely indicates that the mechanical percolation network of Al_2_O_3_ spherical NPs within the PA12 and PLA thermoplastic polymeric matrices is achieved above 1.0 and up to 2.0 wt.%. Several plausible mechanisms have been reported that might lead to mechanical reinforcement of 3DP nanocomposite polymeric materials, especially strength and stiffness, which are amongst others: (i) the optimum polymer melt rheology and temperature during 3DP melt processing, (ii) the strong polymer matrix-filler interaction, (iii) an adequate and high-quality of nanoparticles’ dispersion in the polymer matrix (percolation), via optimum filament compounding-melt mixing extrusion process, and (iv) the size and geometry of the filler as well as its surface chemistry [77]. In our study, it can be concluded that above 2.0 wt.%, the tensile properties for both the filaments as well as the 3DP specimens are not any more affected, while higher filler loadings cannot induce any positive reinforcement mechanism. As such, one should focus in the range of 1.0–2.0 wt.%, since higher filler loadings can result in a knock-down mechanism for the nanocomposites’ mechanical properties, i.e., due to less polymer chain mobility in the melt state that can affect 3DP processability and specimens’ interlayer fusion [78,79], possible nanoparticle agglomeration due to their extremely high surface area [80], which might lead to stress concentration upon mechanical loading [81], etc.

### 3.7. SEM Morphological Analysis of the Tensile Test Specimen Fractured Surfaces

Fractography analyses represented by the corresponding SEM images of tensile specimen fractured surfaces are presented in Figure 14 (PA12 nanocomposites) and Figure 15 (PLA nanocomposites). For the PA12/Al_2_O_3_ nanocomposites (Figure 14), in all cases apart from the PA12/Al_2_O_3_ (1.0 wt.%) (Figure 14A), a relatively ductile fracture mechanism can be observed with rough fractured surfaces and 3DP filaments in the fractured surface. However, no distinct filaments from the different additively manufactured/deposited layers can be observed, neither interlayer nor intralayer voids, being a sign of high-quality interlayer fusion and optimum 3DP manufacturing parameters selected in this study. For the PLA/Al_2_O_3_ nanocomposites (Figure 15), in all cases, a relatively brittle fracture mechanism can be seen with some typical morphology of polymeric materials’ brittle fracturing. Again, the high quality of the sample’s cross-section/fracture surface can be observed without visible intralayer voids (apart from some voids appearing in PLA/Al_2_O_3_ (4.0 wt.% specimen), indicating the optimum 3DP parameters which have been selected for PLA and PLA/Al_2_O_3_ nanocomposites in this study.

### 3.8. Flexural Tests Results of 3D Printed Pure PA12, PLA, and Their Al_2_O_3_ Nanocomposites

Figure 16A,B show the comparative and representative flexural stress (MPa) vs. strain (%) curves for PA12 and PA12/Al_2_O_3_ nanocomposites, as well as for PLA and PLA/Al_2_O_3_ nanocomposites, respectively (all curves for 3DP specimens described by the specific ASTM, with all details given in the Experimental section). Figure 16C,D summarize the flexural strength and flexural moduli values for all the different 3DP specimens (average values and standard deviation). For all specimens, namely, PA12 and PLA nanocomposites, Al_2_O_3_ NPs endowed a prominent reinforcement mechanism for the flexural properties, as previously observed for the case of tensile properties comprehensively discussed in the previous paragraph. Specifically, the highest increase has been experimentally determined for flexural strength and modulus at 1.0 wt.% for PA12 (32.3% and 40.9%, respectively), while at 2.0 wt.% for PLA (35.9% and 34.8%, respectively). It is worth mentioning that the flexural properties of 3DP specimens agree with the properties determined in the tensile experiments, in terms of nanoparticle filler loading and trend in the increase of flexural modulus and strength properties for PA12 and PLA nanocomposites. In the flexural tests, a clear reinforcement is observed on the polymers with the addition of alumina. Even the lowest flexural strength values reported are higher than the corresponding values of the pure polymers for concentrations up to 4 wt.%, which were studied in the work.

### 3.9. Tensile-Flexural Toughness, Impact Strength, and Micro-Hardness Properties

In Figure 17, the tensile toughness (Figure 17A), flexural toughness (Figure 17B), impact strength (Figure 17C), and micro-hardness (Vickers (HV)) (Figure 17D) properties of neat PA12 and PLA, as well as of their respective nanocomposites at 1.0, 2.0, 3.0, and 4.0 wt.% of Al_2_O_3_ filler loading are presented. More specifically, the tensile toughness has shown its maximum increase for the 4.0 wt.% nanocomposites, namely 40.8% for the PA12/Al_2_O_3_ (4.0 wt.%) and 35.6% for the PLA/Al_2_O_3_ (4.0 wt.%) nanocomposite. The flexural toughness was maximum for PA12/Al_2_O_3_ (1.0 wt.%) with a 24.9% increase and PLA/Al_2_O_3_ (2.0 wt.%) with a 31.6% increase. For all filler loadings, the tensile and flexural toughness has been increased indicating that the Al_2_O_3_ nanoparticles’ existence increased the material’s resistance to fracture via inhibition of some crack initiation or crack growth and propagation upon applying a quasi-static tensile or flexural mechanical stress field. The impact strength (Figure 17C) has shown the best performance for PA12/Al_2_O_3_ (3.0 wt.%) with a 30.1% increase, and PLA/Al_2_O_3_ (4.0 wt.%) with a 30.2% increase, while for micro-hardness it has been the same trend, namely for PA12/Al_2_O_3_ (3.0 wt.%) 27.6% increase, and PLA/Al_2_O_3_ (4.0 wt.%) 39.0% increase. The results are quite reasonable for the toughness-related, impact, and micro-hardness properties, which exhibit a trend to be higher and more positively affected with the increased filler loading. This has to do as previously mentioned with some crack-related mechanism that allows the material to “absorb” a higher amount of mechanical energy up to the fracture point, even if the pure tensile and flexural properties’ performance (strength, modulus) had been found to be affected by a generated/achieved mechanical percolation threshold in the range between 1.0 and 2.0 wt.% of filler loading.

## 4. Conclusions

In this work, PA12 and PLA nanocomposite 3DP filaments have been produced via versatile and scalable melt–mixing/compounding processes, with the aim to be used in FFF 3DP as feedstock materials and improve the mechanical properties of the 3DP-built manufactured specimens. The aim of the work was achieved, and alumina improved the mechanical response of the prepared nanocomposites in all concentrations studied.

Different filler Al_2_O_3_ loadings, namely 1.0, 2.0, 3.0, and 4.0 wt.% have been incorporated in PA12 and PLA, respectively, as two polymeric materials widely used as thermoplastic matrices in 3DP with a high potential towards a great variety of engineered applications, e.g., mechanical parts in machines, interior parts in automotive, marine, etc. Next, the developed filaments were used to manufacture 3DP standard samples in geometries according to different ASTM protocols, and an extensive mechanical properties test campaign has been followed to investigate the effect of the Al_2_O_3_ nanofillers in the mechanical reinforcement of PA12 and PLA, respectively. Specifically, to assess the response of the developed 3DP nanocomposites to nano-filler loading, the 3DP samples were comprehensively characterized via mechanical (tensile, flexural), impact, micro-hardness, physicochemical, and fractographic analyses. As a summary, the overall mechanical properties of the 3DP, PA12 and PLA nanocomposites investigated in this study are illustrated in the histogram of Figure 18.

It can be observed that the incorporation of Al_2_O_3_ nanoparticles in both polymeric matrices improves the mechanical properties of the neat polymer for concentrations up to 2 wt.%. Only the toughness values, the impact strength, and the microhardness for the different nanocomposites are found to be more positively affected by further increasing the nanoparticle filler loadings, especially at 3.0 and 4.0 wt.%, which is more possibly explained as previously mentioned due to some crack related mechanism, i.e., the highest filler loading allows the material to “absorb” a higher amount of mechanical energy up to the fracture point, even if the pure tensile and flexural properties performance (strength, modulus) have been found to be affected by a generated/achieved mechanical percolation threshold found in the range between 1.0 and 2.0 wt.% of filler loading. It can be deduced that commercially available PA12 and PLA reinforced with Al_2_O_3_ nanoparticles could be a viable solution for engineering applications where strength, stiffness, toughness, impact, and microhardness properties are critical to be enhanced.

The effect of alumina on two popular thermoplastics in FFF 3D printing (PA12 and PLA) was investigated and determined within the contents of this work. Based on the results of the work, in future work, binary inclusions using alumina as one of the fillers can now be investigated for the development of multi-functional nanocomposites in FFF 3D printing. Additional properties of the nanocomposites investigated herein can also be studied, such as electrical properties, optical properties, and others.

## Figures and Tables

**Figure 1 nanomaterials-12-04292-f001:**
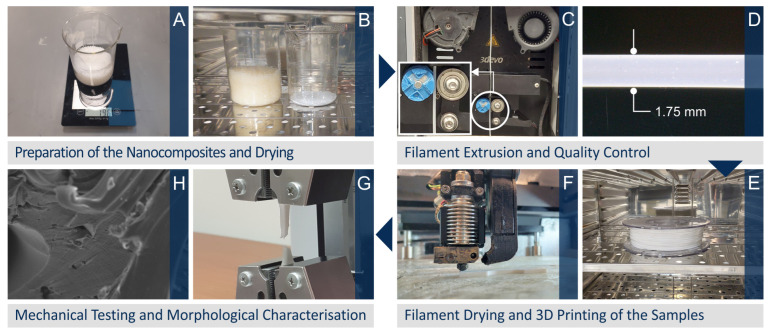
Steps of the methodology followed for the preparation of the 3DP filaments and the 3DP specimens towards the final testing and characterization (**A**) weight of the raw materials, (**B**) raw materials drying process, (**C**) filament extrusion, (**D**) filament inspection, (**E**) filament drying, (**F**) 3D printing of the samples, (**G**) mechanical characterization (a screenshot from a tensile test is shown in the picture), (**H**) morphological characterization with Scanning Electron Microscopy (SEM).

**Figure 2 nanomaterials-12-04292-f002:**
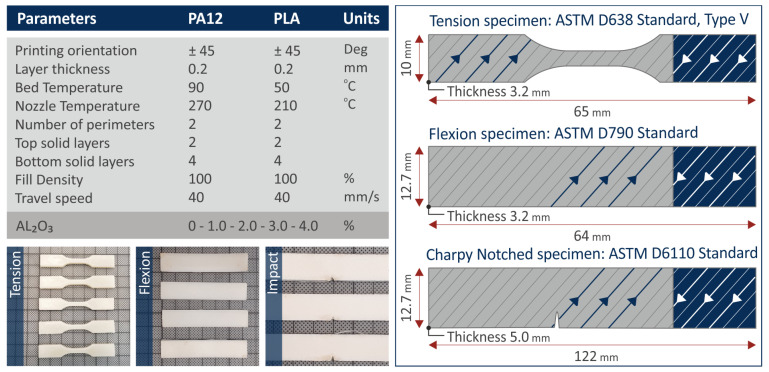
The optimal FFF 3DP parameters set up to the slicer software for the manufacturing of the neat PA12 and PLA and the Al_2_O_3_ nanocomposites (real representative tension, flexural, and impact specimens are depicted).

**Figure 3 nanomaterials-12-04292-f003:**
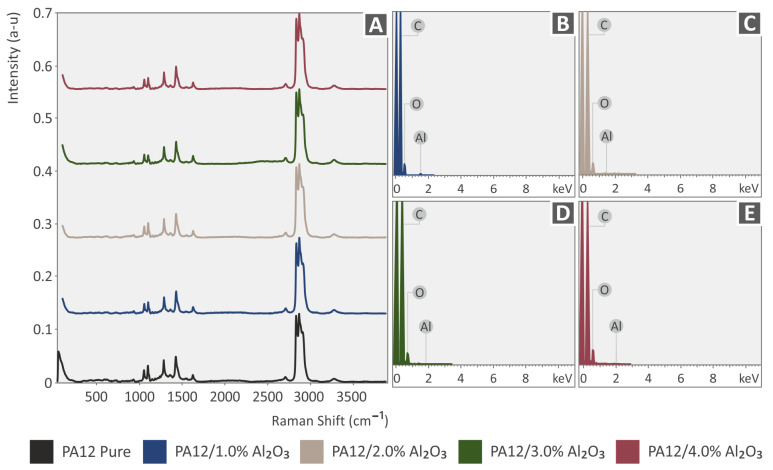
(**A**) Raman spectra for neat PA12, PA12 with 1.0 wt.% Al_2_O_3_, PA12 with 2.0 wt.% Al_2_O_3_, PA12 with 3.0 wt.% Al_2_O_3_, and (**B**–**E**) EDS spectra for all 3DP PA12/Al_2_O_3_ nanocomposites showing the existence of atomic Al arising from the Al_2_O_3_ incorporated nanoparticles in the PA12 polymer matrix.

**Figure 4 nanomaterials-12-04292-f004:**
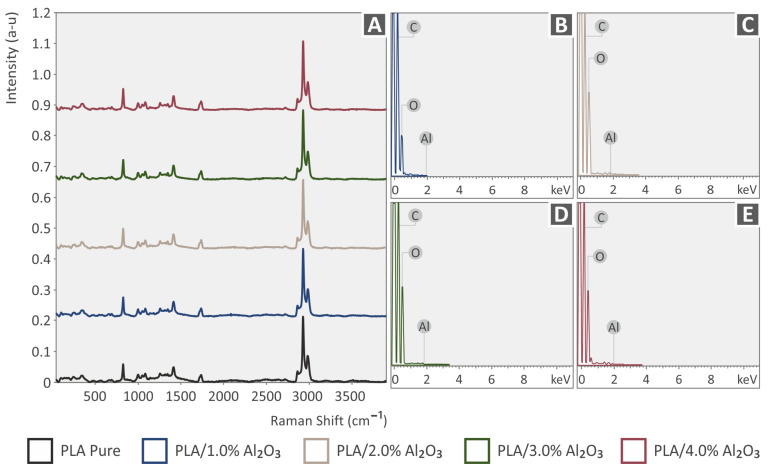
**(A)** Raman spectra for neat PLA, PLA with 1.0 wt.% Al_2_O_3_, PLA with 2.0 wt.% Al_2_O_3_, PLA with 3.0 wt.% Al_2_O_3_, and (**B**–**E**) EDS spectra for all 3DP PLA/Al_2_O_3_ nanocomposites showing the existence of atomic Al arising from the Al_2_O_3_ incorporated nanoparticles in the PLA polymer matrix.

**Figure 5 nanomaterials-12-04292-f005:**
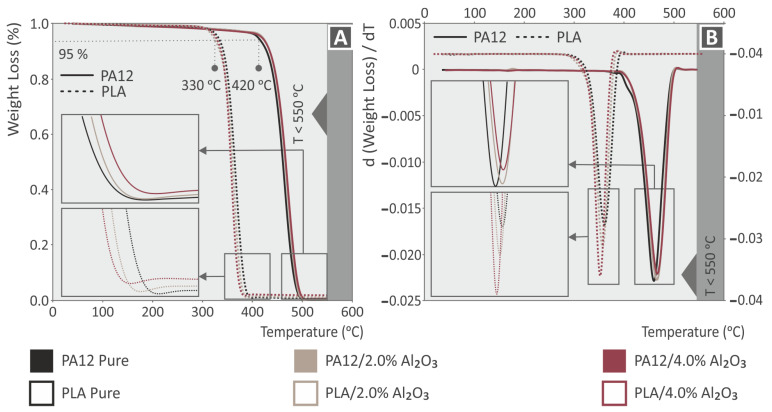
(**A**) Thermogravimetric analysis (TGA), and (**B**) Differential thermogravimetry (DTG) plots of 3DP printed PA12, PLA, and their Al_2_O_3_ nanocomposites at 1.0, 2.0, 3.0, and 4.0 wt.% filler loadings.

**Figure 6 nanomaterials-12-04292-f006:**
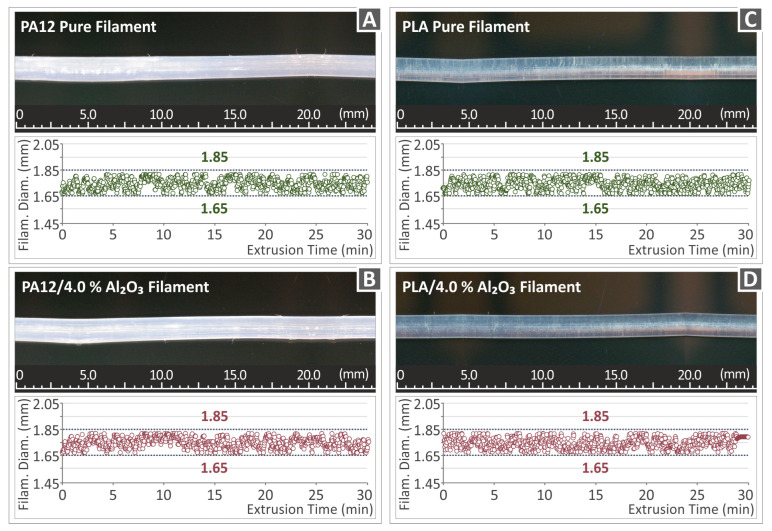
Real-time monitored filament diameter metrology of (**A**,**C**) neat PA12, (**B**,**D**) PA12/Al_2_O_3_ (4.0 wt.%). (**A**,**B**) are optical microscope images, (**C**,**D**) are stereoscope images.

**Figure 7 nanomaterials-12-04292-f007:**
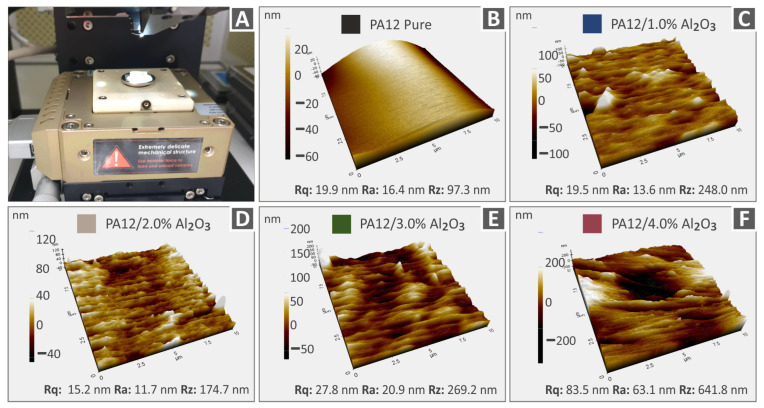
AFM topography images together with the corresponding roughness values (Rq, Ra, Rz) of PA12 and PA12/Al_2_O_3_ nanocomposite filaments (**A**) AFM setup, (**B**) PA12 pure, (**C**) PA12/Al_2_O_3_ 1.0 wt.%, (**D**) PA12/Al_2_O_3_ 2.0 wt.%, (**E**) PA12/Al_2_O_3_ 3.0 wt.%, (**F**) PA12/Al_2_O_3_ 4.0 wt.%.

**Figure 8 nanomaterials-12-04292-f008:**
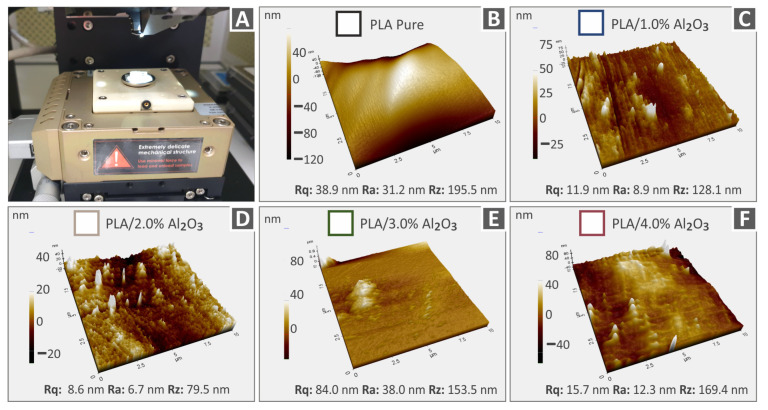
AFM topography images together with the corresponding roughness values (Rq, Ra, Rz) of PLA and PLA/Al_2_O_3_ nanocomposite filaments (**A**) AFM setup, (**B**) PLA pure, (**C**) PLA/Al_2_O_3_ 1.0 wt.%, (**D**) PLA/Al_2_O_3_ 2.0 wt.%, (**E**) PLA/Al_2_O_3_ 3.0 wt.%, (**F**) PLA/Al_2_O_3_ 4.0 wt.%.

**Figure 9 nanomaterials-12-04292-f009:**
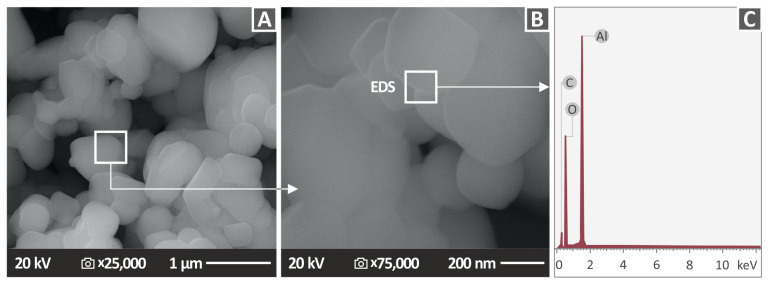
SEM images of Al_2_O_3_ powder at two magnifications: (**A**) 25,000×, (**B**) 75,000×, and (**C**) the corresponding EDS spectrum acquired from the Al_2_O_3_ powder.

**Figure 10 nanomaterials-12-04292-f010:**
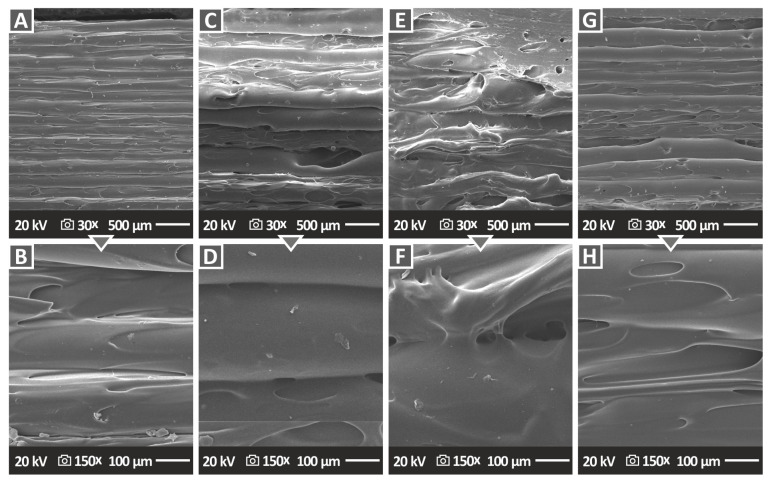
SEM images of the side surface morphology of (**A**,**B**) PA12/Al_2_O_3_ (1.0 wt.%), (**C**,**D**) PA12/Al_2_O_3_ (2.0 wt.%), (**E**,**F**) PA12/Al_2_O_3_ (3.0 wt.%), and (**G**,**H**) PA12/Al_2_O_3_ (4.0 wt.%).

**Figure 11 nanomaterials-12-04292-f011:**
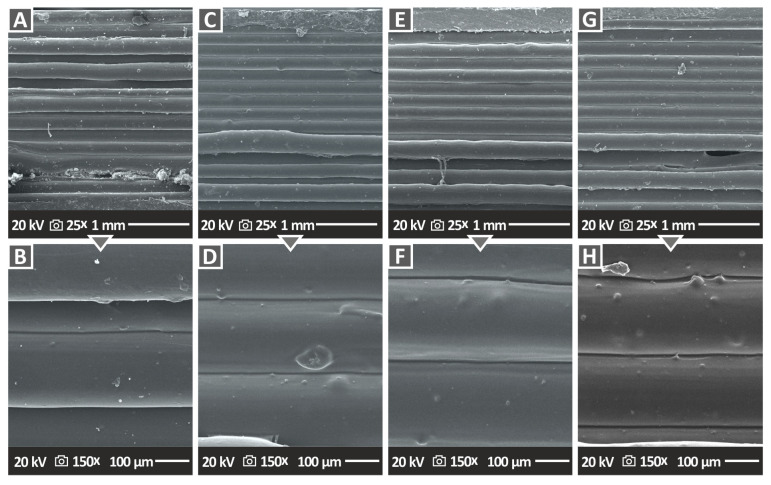
SEM images of the side surface morphology of (**A**,**B**) PLA/Al_2_O_3_ (1.0 wt.%), (**C**,**D**) PLA/Al_2_O_3_ (2.0 wt.%), (**E**,**F**) PLA/Al_2_O_3_ (3.0 wt.%), and (**G**,**H**) PLA/Al_2_O_3_ (4.0 wt.%).

**Figure 12 nanomaterials-12-04292-f012:**
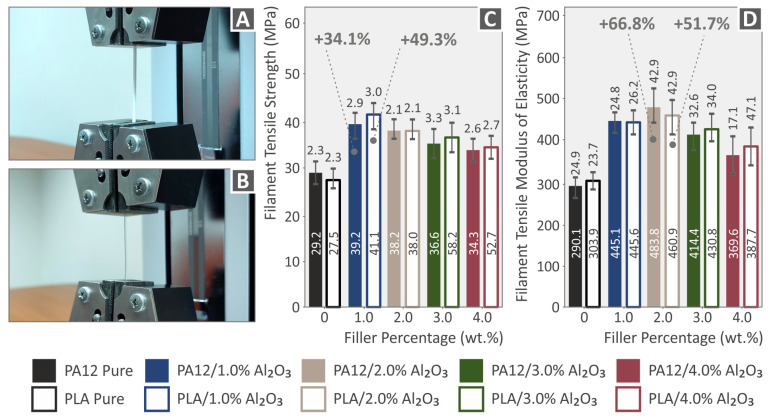
(**A**,**B**) Representative neat polymer and nanocomposite polymer filament under testing; (**C**,**D**) Tensile strength and tensile modulus of elasticity for 3DP PA12, PLA, and all nanocomposite filaments at 1.0, 2.0, 3.0, and 4.0 wt.% Al_2_O_3_ filler loading. Legend shows which color corresponds to which material in each graph.

**Figure 13 nanomaterials-12-04292-f013:**
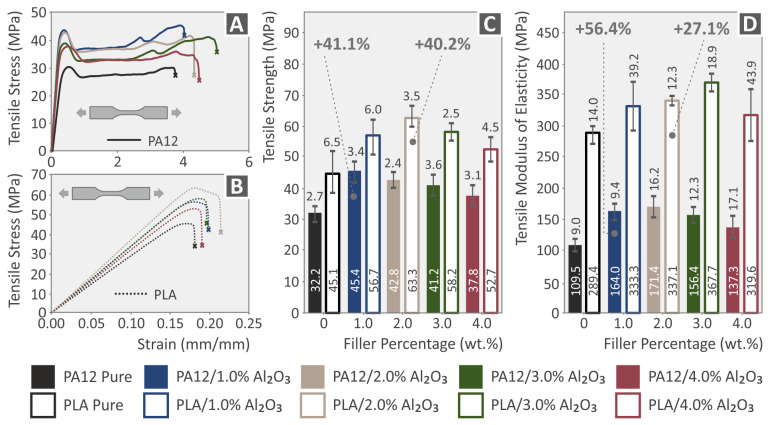
(**A**,**B**) Tensile stress (MPa) vs. strain (%) stress–strain curves for PA12 and PA12/Al_2_O_3_ nanocomposites, as well as for PLA and PLA/Al_2_O_3_ nanocomposites; (**C**,**D**) Tensile strength and tensile modulus of elasticity for 3DP PA12, PLA and all nanocomposites at 1.0, 2.0, 3.0, and 4.0 wt.% Al_2_O_3_ filler loading. Legend shows which color corresponds to which material in each graph.

**Figure 14 nanomaterials-12-04292-f014:**
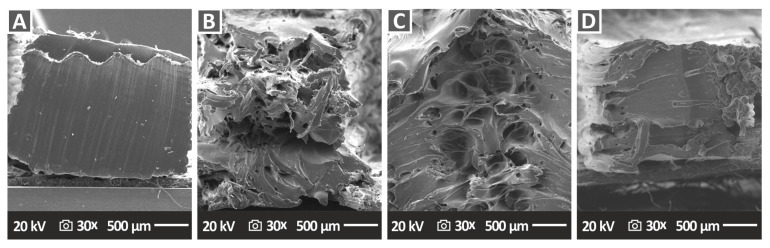
SEM images of the tensile specimens’ fractured surface morphology of (**A**) PA12/Al_2_O_3_ (1.0 wt.%), (**B**) PA12/Al_2_O_3_ (2.0 wt.%), (**C**) PA12/Al_2_O_3_ (3.0 wt.%), and (**D**) PA12/Al_2_O_3_ (4.0 wt.%), all at 30× magnification.

**Figure 15 nanomaterials-12-04292-f015:**
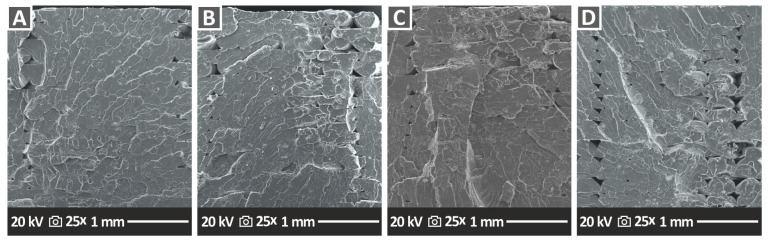
SEM images of the tensile specimens’ fractured surface morphology of (**A**) PLA/Al_2_O_3_ (1.0 wt.%), (**B**) PLA/Al_2_O_3_ (2.0 wt.%), (**C**) PLA/Al_2_O_3_ (3.0 wt.%), and (**D**) PLA/Al_2_O_3_ (4.0 wt.%), all at 25× magnification.

**Figure 16 nanomaterials-12-04292-f016:**
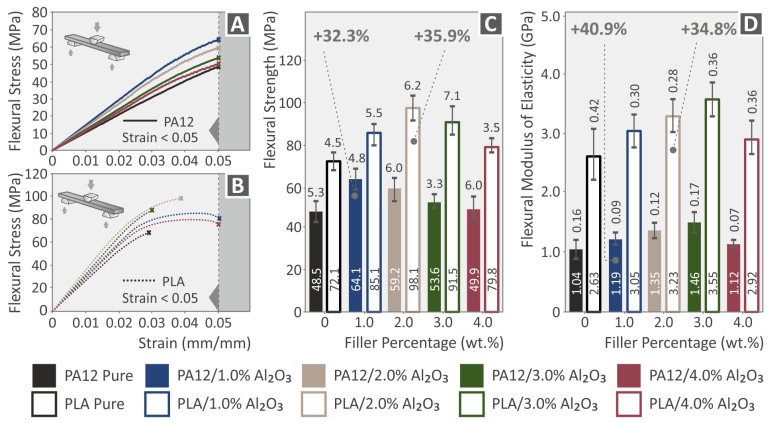
(**A**,**B**) Flexural stress (MPa) vs. strain (%) comparative and representative stress–strain curves for PA12 and PA12/Al_2_O_3_ nanocomposites, as well as for PLA and PLA/Al_2_O_3_ nanocomposites; (**C**,**D**) Average flexural strength and flexural modulus, along with the calculated standard deviation values, for 3DP PA12, PLA and all nanocomposites at 1.0, 2.0, 3.0, and 4.0 wt.% Al_2_O_3_ filler loading. Legend shows which color corresponds to which material in each graph.

**Figure 17 nanomaterials-12-04292-f017:**
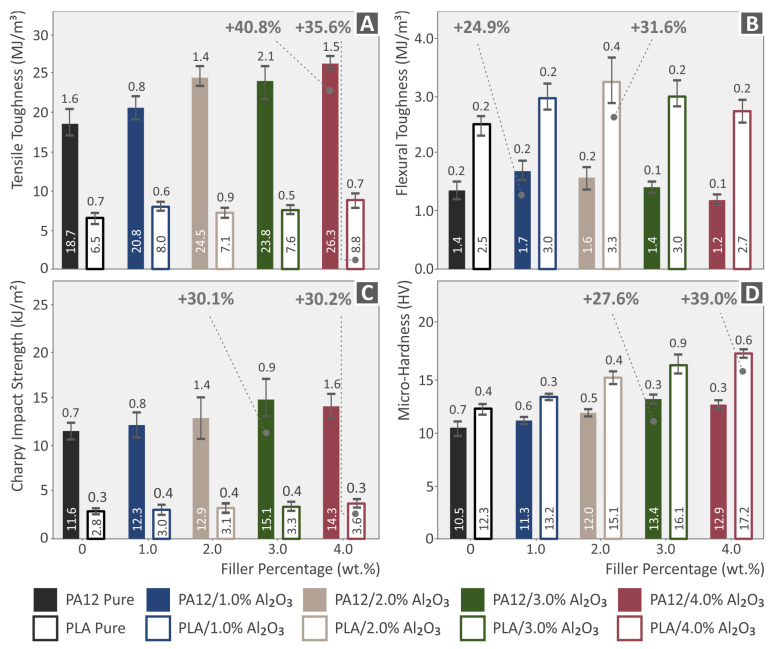
Tensile toughness (**A**), Flexural toughness (**B**), Charpy’s notched impact strength (kJ/m^2^) (**C**), and Micro-hardness (Vickers (HV)) (**D**) properties of neat PA12 and PLA, as well as of their respective nanocomposites at 1.0, 2.0, 3.0, and 4.0 wt.% of Al_2_O_3_ filler loading. Legend shows which color corresponds to which material in each graph.

**Figure 18 nanomaterials-12-04292-f018:**
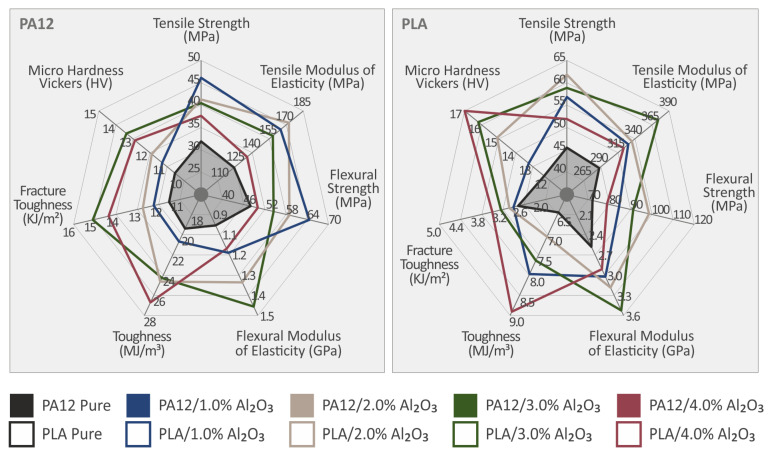
Spider graph summarizing the experimentally determined mechanical properties of pure PA12 and PLA compared to PA12/Al_2_O_3_ and PLA/Al_2_O_3_ 3DP nanocomposites at 1.0, 2.0, 3.0, and 4.0 wt.% of Al_2_O_3_ filler loading. The shaded area presents the mechanical response of the pure materials. Legend shows which color corresponds to which material in each graph.

**Table 1 nanomaterials-12-04292-t001:** Major Raman peaks identified and their related assignments.

Wavenumber (cm^−1^)	Raman Peak Assignment
1060	C–O–C stretching [68]
1105	C–O–C stretching [68]
1293	C–O–C stretching [68]
1434	CH_2_ deformation [68,69,70]
2850	CH_2_ symmetric stretching [70]
2884	CH_2_ symmetric stretching [70]
2923	CH_2_ asymmetric stretching [70]

**Table 2 nanomaterials-12-04292-t002:** Major Raman peaks identified and their related assignments.

Wavenumber (cm^−1^)	Raman Peak Assignment
870	C–COO stretching [71]
1115	CH_3_ rocking [71]
1374	C–H bending [72]
1449	CH_3_ bending [71]
1761	C=O stretching [71]
2888	C–H antisymmetric stretching [73]
2945	C–H stretching and bending [72]
2996	CH_3_ asymmetric stretch [73]

## Data Availability

Data from the research will be available upon request.

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
