# Peer review of "Three-Dimensional Printed Polyamide 12 (PA12) and Polylactic Acid (PLA) Alumina (Al2O3) Nanocomposites with Significantly Enhanced Tensile, Flexural, and Impact Properties"

_nanomaterials, 2022, doi:10.3390/nano12234292_

Round 1

Reviewer 1 Report

In this work, Petousis et al. use polyamide12 (PA12) and polylactic acid (PLA) to create 3D printed composites with AL2O3. The polymer/AL2O3 mixtures and the 3D printed composites are well characterized through various techniques. The mechanical property results for the 3D printed parts are interesting, as they show the effect of loading on different tensile properties. Overall the work is conducted well and I have some minor comments below:

11.       What is the possible explanation for PLA composite showing less roughness than the PA12 composites?

22.       Does EDS show the expected 2:3 ratio for Al and O? Also, what impurities are in these NPs that allows carbon to be seen from the EDS data?

33.       The top half of Fig 11 shows 25x magnification, but the text says 30x.

44.       Is the sample in Fig 12 black in color? If so, this indicates significant carbon contamination in the Al2O3 (also from the carbon peak in EDS). Is there any idea what form this carbon is in, what amount, and if it plays a role in the composite properties?

Author Response

Comments from Reviewer 1

Answers of the Authors

In this work, Petousis et al. use polyamide12 (PA12) and polylactic acid (PLA) to create 3D printed composites with AL2O3. The polymer/AL2O3 mixtures and the 3D printed composites are well characterized through various techniques. The mechanical property results for the 3D printed parts are interesting, as they show the effect of loading on different tensile properties. Overall the work is conducted well and I have some minor comments below:

The authors would like to thank the reviewer for the kind words, the constructive comments, and the effort toward the improvement of our work. The authors hope that the reviewer will find the revised version of the manuscript improved according to the proposed, reviewer’s comments, and instructions.

11.       What is the possible explanation for PLA composite showing less roughness than the PA12 composites?

The authors agree with the reviewer. Measured surface roughness values are very low and differences overall are not significant, still this is something worth discussing. Any differences can be attributed to the different rheological properties of the studied thermoplastics, which affect the surface structure of the materials. The addition of the Aluminum oxide filler has different effect in the structure of each thermoplastic. Additionally, measurements were taken at random positions, so differences are expected also due to the topography of the microscale region the measurements were taken. This is now discussed in the revised version of the work.

22.       Does EDS show the expected 2:3 ratio for Al and O? Also, what impurities are in these NPs that allows carbon to be seen from the EDS data?

The authors would like to thank the reviewer for the comment. The thermoelastic materials are organic materials. The following work was added to support this:

Maréchal, E. (2005). Creation and Development of Thermoplastic Elastomers, and Their Position among Organic Materials. In Handbook of Condensation Thermoplastic Elastomers, S. Fakirov (Ed.). https://doi.org/10.1002/3527606610.ch1

So, the presence of Carbon in the EDS is expected. Additionally, after the sputtering process, the sample are washed, before they are entered in the SEM chamber, and traces from organic structures can remain in them. If the reviewer is referring to the EDS graph on the alumina powder and not on the 3D printed samples, the powder is placed in a carbon tape, to be inspected with SEM, so the presence of carbon is expected.

Regarding the ratio for Al and O, EDS is an indicative process regarding the presence of the elements, it is not for precise stoichiometric ratio or precise quantification of the concentration of an element since measurements are taken in small region. High peaks are indicative for high concentration of an element in the region under observation, but this is mainly for qualitative assessment not a quantitative one. This is now discussed in the work in the revised version of the manuscript.

33.       The top half of Fig 11 shows 25x magnification, but the text says 30x.

The authors would like to thank the reviewer for the comment. This is now corrected in the revised version of the manuscript.

44.       Is the sample in Fig 12 black in color? If so, this indicates significant carbon contamination in the Al2O3 (also from the carbon peak in EDS). Is there any idea what form this carbon is in, what amount, and if it plays a role in the composite properties?

The authors would like to thank the reviewer for the comment. This is now corrected in the revised version of the manuscript. This image was added by mistake. The correct colors of the pure filament and the filament with the Al2O3 additive are shown in figure 6. Thank you.

Reviewer 2 Report

Comment 1:  The work is novel and exciting for readers, but the abstract is not eye-catching, so I suggest rewriting the abstract. Add the main quantified results to the Abstract. The Abstract should be rewritten better to include novelty, methodology, and main results.

Comment 2: A recapitulation of previous works is shown in the introduction. However, it would be desirable to have a more detailed discussion about the primary outcomes of those works. For example, answering some of the following questions may help to get more explicit content: Why are these works relevant? Which practical problems were addressed? Do you know how previous results are related to the proposed work? What are the outstanding, unresolved research questions? Please add some practical examples to illustrate the points of interest. The authors include many time group citations, so I strongly recommend writing each article separately and writing what the authors did and what was the main finding.

Comment 3: More related references should be added to the introduction section. Must emphasize the literature on Polyamide 12 (PA12) and Polylactic 2 acid (PLA) alumina (Al2O3) nanocomposites. 

Comment 4: Please expand the motivation, and the problem context, clarify the problem description, and (if possible) add specific objectives. For validation of governing equations, one must cite some references and a few references in the boundary conditions.

Comment 5A large number of terminologies and symbols are used throughout the text. A nomenclature table is needed. 

Comment 6: In “Results and discussion,” it is suggested to discuss the proposed results' advantages. More discussions are needed on the trends perceived in all the figures. 

Comment 7: Please expand the conclusions about the specific goals and future work.

Author Response

Comments from Reviewer 2

Answers of the Authors

Comment 1: The work is novel and exciting for readers, but the abstract is not eye-catching, so I suggest rewriting the abstract. Add the main quantified results to the Abstract. The Abstract should be rewritten better to include novelty, methodology, and main results.

The authors would like to thank the reviewer for the constructive comments, and the effort toward the improvement of our work. The authors hope that the reviewer will find the revised version of the manuscript improved according to the proposed, reviewer’s comments, and instructions.

Following the reviewer’s comment, the abstract section was amended in the revised version of the manuscript.

Comment 2: A recapitulation of previous works is shown in the introduction. However, it would be desirable to have a more detailed discussion about the primary outcomes of those works. For example, answering some of the following questions may help to get more explicit content: Why are these works relevant? Which practical problems were addressed? Do you know how previous results are related to the proposed work? What are the outstanding, unresolved research questions? Please add some practical examples to illustrate the points of interest. The authors include many time group citations, so I strongly recommend writing each article separately and writing what the authors did and what was the main finding.

The authors would like to thank the reviewer for the comment. The literature in the work is now more analytically presented in the revised version of the manuscript.

Comment 3: More related references should be added to the introduction section. Must emphasize the literature on Polyamide 12 (PA12) and Polylactic 2 acid (PLA) alumina (Al2O3) nanocomposites. 

The authors would like to thank the reviewer for the comment. Following the reviewer’s comment literature on Polyamide 12 (PA12) and Polylactic 2 acid (PLA) alumina (Al2O3) nanocomposites were added in the revised version of the manuscript. More specifically, the following works were added in the revised version of the manuscript:

1.          Jo, H.Y.; Jung, D.S.; Lee, S.-H.; Kim, D.S.; Lee, Y.K.; Lim, H.M. Characterization of Composites Prepared with Polyamide-Imide and Alumina Synthesized by Solvothermal Method. Nanosci. Nanotechnol. Lett. 2016, 8.

2.          Oliveira, I. de; Vernilli, F.; Vieira, R.; da Silva, J.V.L. Obtainment of the alumina polyamide-coated spherical particles. Powder Technol. 2019, 356, 112–118, doi:https://doi.org/10.1016/j.powtec.2019.08.003.

3.            Nakonieczny, D.S.; Kern, F.; Dufner, L.; Antonowicz, M.; Matus, K. Alumina and Zirconia-Reinforced Polyamide PA-12 Composites for Biomedical Additive Manufacturing. Materials (Basel). 2021, 14.

4.          Bouamer, A.; Younes, A. Effect of ZnO, SiO2 and Al2O3 Doped on Morphological, Optical, Structural and Mechanical Properties of Polylactic Acid. Key Eng. Mater. 2022, 911, 105–113, doi:10.4028/p-79tw2s.

5.          Jiang, J.; Yang, S.; Li, L.; Bai, S. High thermal conductivity polylactic acid composite for 3D printing: Synergistic effect of graphene and alumina. Polym. Adv. Technol. 2020, 31, 1291–1299, doi:https://doi.org/10.1002/pat.4858.

6.          Chen, J.; Hu, R.-R.; Jin, F.-L.; Park, S.-J. Synergistic reinforcing of poly(lactic acid) by poly(butylene adipate-co-terephthalate) and alumina nanoparticles. J. Appl. Polym. Sci. 2021, 138, 50250, doi:https://doi.org/10.1002/app.50250.

7.          Kurtycz, P.; Karwowska, E.; Ciach, T.; Olszyna, A.; Kunicki, A. Biodegradable polylactide (PLA) fiber mats containing Al2O3-Ag nanopowder prepared by electrospinning technique — Antibacterial properties. Fibers Polym. 2013, 14, 1248–1253, doi:10.1007/s12221-013-1248-3.

8.            Mujeeb, A.; Lobo, A.G.; Antony, A.J.; Ramis, M.K. An Experimental Study on the Thermal Properties and Electrical Properties of Polylactide Doped with Nano Aluminium Oxide and Nano Cupric Oxide. Ina. Lett. 2017, 2, 145–151, doi:10.1007/s41403-017-0030-z.

Comment 4Please expand the motivation, and the problem context, clarify the problem description, and (if possible) add specific objectives. For validation of governing equations, one must cite some references and a few references in the boundary conditions.

The authors would like to thank the reviewer for the comment. The motivation and the problem context and the objectives of the work are now more clearly presented in the revised version of the manuscript.

Comment 5: A large number of terminologies and symbols are used throughout the text. A nomenclature table is needed. 

The authors would like to thank the reviewer for the comment. Following the reviewer’s comment, a nomenclature was added in the revised version of the manuscript.

Comment 6: In “Results and discussion,” it is suggested to discuss the proposed results' advantages. More discussions are needed on the trends perceived in all the figures. 

The authors would like to thank the reviewer for the comment. The results of the work and the trends perceived in all the figures are now more analytically presented in the revised version of the manuscript.

Comment 7: Please expand the conclusions about the specific goals and future work.

The authors would like to thank the reviewer for the comment. The conclusions section is now revised, and the goals of the work are more clearly presented. The future work is also now mentioned. Now the effect on alumina on two popular thermoplastics in MEX 3D printing is investigated. Based on the results of the work, binary inclusions using alumina as one of the fillers can now be investigated, toward the development of multi-functional nanocomposites. Additional properties of the nanocomposites investigated herein can also be studied, such as electrical properties, optical properties and other.

Reviewer 3 Report

Dear Authors,

The article is very good, factually and very concisely written. Something cu arouses my admiration in detail is the very good methodology, in an exemplary way planned experiment and the presentation of the results leaving no room for negative interpretation from the reviewer. For me, as a materials and process chemist, the presentation of results from RS and TG is particularly important. You have done it very well - especially the analysis of the spectra. I recommend the article for acceptance in MDPI POLYMERS after minor corrections and clarification of some of my questions - because I will frankly admit your article made me very curious and I would like to learn more on this topic ;) 

Strong Points

- a very well-developed methodology,

- very good presentation of results - especially RS and mechanical tests (basically the whole set except adhesion), realy good job! 

- graphically the article at a very high level, it reflects the quality of the results obtained.

Weakness

- your research is not the first in this area as you claim; there are papers (from a year ago and from this year) from one author who moved exactly the same subject (I give the sources below) additionally supplemented with a new printing method and explanation of mass filler shares: 

(1) Alumina and Zirconia-Reinforced Polyamide PA-12 Composites for Biomedical Additive Manufacturing

(2) Effect of Calcination Temperature on the Phase Composition, Morphology, and Thermal Properties of ZrO2 and Al2O3 Modified with APTES (3-aminopropyltriethoxysilane)

(3) PA-12-Zirconia-Alumina-Cenospheres 3D Printed Composites: Accelerated Ageing and Role of the Sterilisation Process for Physicochemical Properties

and here in this regard to your intro it would be worthwhile to introduce additional information:

- why there is a problem with fillers in composites - especially with FDM(FFF),

- what is the problem of phase seperation in composites and what is the role of coupling agents (a well-known problem, for example, in injection molding and in dentistry in polymers with silica),

- why there is a problem with homogeneous dispersion of fillers in composites, how it relates to pre-compounding in FDM and why there is a problem with dosing of fillers during preparation of the filament

- and most importantly - the problem of wettability of the filler surface by the liquid polymer...this is a major challenge and the basis for achieving good adhesion - here it is all about functional groups (especially those containing nitrogen and carboxyl)

You will find the answers to these questions in the first three publications above and the ones I give you below, read them, make changes to your intro and quote them:

(1) Assessment of some characteristics, properties of a novel waterborne acrylic coating incorporated TiO2 nanoparticles modified with silane coupling agent and Ag/Zn zeolite

(2) Enhanced thermo-mechanical properties of acrylic resin reinforced with silanized alumina whiskers

(3) 3D printed ceramic slurries with improved solid content through optimization of alumina powder and coupling agent

(4) 3D printed ceramic slurries with improved solid content through optimization ofRealization of complex-shaped and high-performance alumina ceramic cutting tools via Vat photopolymerization based 3D printing: A novel surface modification strategy through coupling agents aluminic acid ester and silane coupling agent alumina powder and coupling agent

in the research part, I have some questions and a few reservations: 

(I) the oxidants are not 50 nm - this is the size of the crystallites - according to the photos (which you yourself have shown) the size of the alumina is submicrometric about 1 um, please correct this because the information may be misleading,

(II) why do you use such small mass shares of alumina? in the text there is no information on this subject, from what does this result? from your experience or preliminary tests? 

(III) why don't you use pre-compounding before extrusion? 

(IV) how was the filler added: as slurry or raw powder? and how - manually dosed or injected into the chamber during extrusion?

(V) why did you do TG only in air? some of the results (especially) for inert water and crystalline water come out better in inert gas - and in your graphs...not even a trace of moisture. Can you explain this? 

(VI) for thermogravimetry it is good to present the graph in the form of DTA/TGA - because then you can see the dynamics of changes (especially endo and exothermic effects, correct it please

(VII) great that you presented the breakthroughs of the samples on the SEM - I only have one comment on the photos: show what detector you used and what was the beam current.

One more question outside of the official review - what program do you use for the charts? They make a very good impression and I would also like to use this in the future ;) Can I ask you for this information? 

Τους θερμότερους χαιρετισμούς μου, σας εύχομαι καλή τύχη και γόνιμη περαιτέρω εργασία!

Χαιρετισμούς

Κριτής

Author Response

Comments from Reviewer 3

Answers of the Authors

Dear Authors,

The article is very good, factually and very concisely written. Something cu arouses my admiration in detail is the very good methodology, in an exemplary way planned experiment and the presentation of the results leaving no room for negative interpretation from the reviewer. For me, as a materials and process chemist, the presentation of results from RS and TG is particularly important. You have done it very well - especially the analysis of the spectra. I recommend the article for acceptance in MDPI POLYMERS after minor corrections and clarification of some of my questions - because I will frankly admit your article made me very curious and I would like to learn more on this topic ;) 

 Strong Points

- a very well-developed methodology,

- very good presentation of results - especially RS and mechanical tests (basically the whole set except adhesion), realy good job! 

- graphically the article at a very high level, it reflects the quality of the results obtained.

The authors would like to thank the reviewer for the kind words, the constructive comments, and the effort toward the improvement of our work. The authors hope that the reviewer will find the revised version of the manuscript improved according to the proposed, reviewer’s comments, and instructions.

Weakness

- your research is not the first in this area as you claim; there are papers (from a year ago and from this year) from one author who moved exactly the same subject (I give the sources below) additionally supplemented with a new printing method and explanation of mass filler shares: 

(1) Alumina and Zirconia-Reinforced Polyamide PA-12 Composites for Biomedical Additive Manufacturing

(2) Effect of Calcination Temperature on the Phase Composition, Morphology, and Thermal Properties of ZrO2 and Al2O3 Modified with APTES (3-aminopropyltriethoxysilane)

(3) PA-12-Zirconia-Alumina-Cenospheres 3D Printed Composites: Accelerated Ageing and Role of the Sterilisation Process for Physicochemical Properties

and here in this regard to your intro it would be worthwhile to introduce additional information:

- why there is a problem with fillers in composites - especially with FDM(FFF),

- what is the problem of phase seperation in composites and what is the role of coupling agents (a well-known problem, for example, in injection molding and in dentistry in polymers with silica),

- why there is a problem with homogeneous dispersion of fillers in composites, how it relates to pre-compounding in FDM and why there is a problem with dosing of fillers during preparation of the filament

- and most importantly - the problem of wettability of the filler surface by the liquid polymer...this is a major challenge and the basis for achieving good adhesion - here it is all about functional groups (especially those containing nitrogen and carboxyl)

You will find the answers to these questions in the first three publications above and the ones I give you below, read them, make changes to your intro and quote them:

(1) Assessment of some characteristics, properties of a novel waterborne acrylic coating incorporated TiO2 nanoparticles modified with silane coupling agent and Ag/Zn zeolite

(2) Enhanced thermo-mechanical properties of acrylic resin reinforced with silanized alumina whiskers

(3) 3D printed ceramic slurries with improved solid content through optimization of alumina powder and coupling agent

(4) Realization of complex-shaped and high-performance alumina ceramic cutting tools via Vat photopolymerization based 3D printing: A novel surface modification strategy through coupling agents aluminic acid ester and silane coupling agent alumina powder and coupling agent

The authors would like to thank the reviewer for the comment. The works suggested by the reviewer are now considered in the revised version of the manuscript. All the interesting topics suggested by the reviewer are now addressed, following the comment of the reviewer.

Additionally, the following works were added in the revised version of the manuscript:

1.          Jo, H.Y.; Jung, D.S.; Lee, S.-H.; Kim, D.S.; Lee, Y.K.; Lim, H.M. Characterization of Composites Prepared with Polyamide-Imide and Alumina Synthesized by Solvothermal Method. Nanosci. Nanotechnol. Lett. 2016, 8.

2.          Oliveira, I. de; Vernilli, F.; Vieira, R.; da Silva, J.V.L. Obtainment of the alumina polyamide-coated spherical particles. Powder Technol. 2019, 356, 112–118, doi:https://doi.org/10.1016/j.powtec.2019.08.003.

3.            Bouamer, A.; Younes, A. Effect of ZnO, SiO2 and Al2O3 Doped on Morphological, Optical, Structural and Mechanical Properties of Polylactic Acid. Key Eng. Mater. 2022, 911, 105–113, doi:10.4028/p-79tw2s.

4.          Jiang, J.; Yang, S.; Li, L.; Bai, S. High thermal conductivity polylactic acid composite for 3D printing: Synergistic effect of graphene and alumina. Polym. Adv. Technol. 2020, 31, 1291–1299, doi:https://doi.org/10.1002/pat.4858.

5.          Chen, J.; Hu, R.-R.; Jin, F.-L.; Park, S.-J. Synergistic reinforcing of poly(lactic acid) by poly(butylene adipate-co-terephthalate) and alumina nanoparticles. J. Appl. Polym. Sci. 2021, 138, 50250, doi:https://doi.org/10.1002/app.50250.

6.          Kurtycz, P.; Karwowska, E.; Ciach, T.; Olszyna, A.; Kunicki, A. Biodegradable polylactide (PLA) fiber mats containing Al2O3-Ag nanopowder prepared by electrospinning technique — Antibacterial properties. Fibers Polym. 2013, 14, 1248–1253, doi:10.1007/s12221-013-1248-3.

7.          Mujeeb, A.; Lobo, A.G.; Antony, A.J.; Ramis, M.K. An Experimental Study on the Thermal Properties and Electrical Properties of Polylactide Doped with Nano Aluminium Oxide and Nano Cupric Oxide. Ina. Lett. 2017, 2, 145–151, doi:10.1007/s41403-017-0030-z.

in the research part, I have some questions and a few reservations: 

(I) the oxidants are not 50 nm - this is the size of the crystallites - according to the photos (which you yourself have shown) the size of the alumina is submicrometric about 1 um, please correct this because the information may be misleading,

The authors would like to thank the reviewer for the comment. By mistake a 50 nm nanoparticle size was mentioned. The alumina grade (NG04SO103) used in the work has a 180 nm particle size according to its manufacturer. The nanoparticles size, as it is shown in Fig. 9 of the work, verify this size. This is now corrected and more analytically presented in the revised version of the manuscript. Thank you.

(II) why do you use such small mass shares of alumina? in the text there is no information on this subject, from what does this result? from your experience or preliminary tests? 

The authors would like to thank the reviewer for the comment. In the work, we were preparing the nanocomposites with increasing loading, fabricating specimens, testing them, evaluating the results, and then we were increasing the filler loading. When the mechanical properties started to decline, we terminated the experimental course. We have prepared nanocomposites from the beginning with the same process and materials. We have conducted all the tests in this nanocomposite from the beginning in the same way we performed the tests in the original work.

In the work we would like to investigate the reinforcement effect of the alumina additive in the two thermoplastics. We checked up to which concentration the mechanical properties were enhanced. Additionally, we didn’t want to test high filler concentrations which would change significantly other parameters and aspects of the polymers, such as their rheology. At the same time, higher filler concentrations would have an effect in the cost of the produced nanocomposites. This is a threshold analysis for the reinforcing effect, keeping the other parameters as close as possible to the properties of the pure polymers. This is now commented in the revised version of the manuscript.

(III) why don't you use pre-compounding before extrusion? 

The authors would like to thank the reviewer for the comment. A pre-compounding procedure is followed in the work, using a high-power mixer. The procedure is now more clearly presented in the revised version of the manuscript.

(IV) how was the filler added: as slurry or raw powder? and how - manually dosed or injected into the chamber during extrusion?

The authors would like to thank the reviewer for the comment. Raw materials were used. They were first dried. Then weighted with a high precision electronic scale and then mixed at the predefined concentrations. A mixture with the matrix material and the additive is prepared for each concentration. This is now more clearly presented in the revised version of the manuscript.

(V) why did you do TG only in air? some of the results (especially) for inert water and crystalline water come out better in inert gas - and in your graphs...not even a trace of moisture. Can you explain this? 

The authors would like to thank the reviewer for the comment. Samples are dried before the TGA measurements. Measurements are taken in nitrogen atmosphere. By mistake it was mentioned that measurements are taken in air. This is now corrected in the revised version of the manuscript.

(VI) for thermogravimetry it is good to present the graph in the form of DTA/TGA - because then you can see the dynamics of changes (especially endo and exothermic effects, correct it please

The authors would like to thank the reviewer for the comment. Figure 5a presents the TGA measurements results with the weight loss vs temperature for all nano compounds studied in the work. Figure 5b presents the DTA results with the weight loss rate vs temperature for all nano compounds studied in the work. This is now more clearly presented in the revised version of the manuscript.

(VII) great that you presented the breakthroughs of the samples on the SEM - I only have one comment on the photos: show what detector you used and what was the beam current.

The authors would like to thank the reviewer for the comment. A Secondary electrons (SE) detector was used for taking the images in the SEM. This is now mentioned in the revised version of the manuscript.

One more question outside of the official review - what program do you use for the charts? They make a very good impression and I would also like to use this in the future ;) Can I ask you for this information? 

The authors would like to thank the reviewer for the comment and the kind words. Different software tools are used for the preliminary analysis of the results, such as MS excel and Matlab. The final compilation of the figures is implemented using the Corel draw software suite. Thank you.

Round 2

Reviewer 2 Report

accept.